# Biomechanical Modulation of Dental Pulp Stem Cell (DPSC) Properties for Soft Tissue Engineering

**DOI:** 10.3390/bioengineering10030323

**Published:** 2023-03-03

**Authors:** Tara Gross, Martin Philipp Dieterle, Kirstin Vach, Markus Joerg Altenburger, Elmar Hellwig, Susanne Proksch

**Affiliations:** 1Department of Operative Dentistry and Periodontology, Center for Dental Medicine, Medical Center—University of Freiburg, Faculty of Medicine, Albert-Ludwigs-University of Freiburg, Hugstetter Straße 55, 79106 Freiburg, Germany; 2G.E.R.N. Research Center for Tissue Replacement, Regeneration and Neogenesis, Medical Center—University of Freiburg, Faculty of Medicine, Albert-Ludwigs-University of Freiburg, Engesserstr. 4, 79108 Freiburg, Germany; 3Division of Oral Biotechnology, Center for Dental Medicine, Medical Center—University of Freiburg, Faculty of Medicine, Albert-Ludwigs-University of Freiburg, Hugstetter Str. 55, 79106 Freiburg, Germany; 4Institute of Medical Biometry and Statistics, Medical Center—University of Freiburg, Faculty of Medicine, Albert-Ludwigs—University of Freiburg, Stefan-Meier-Str. 26, 79104 Freiburg, Germany; 5Dental Clinic 1–Operative Dentistry and Periodontology, University Hospital Erlangen, Friedrich-Alexander-Universität Erlangen-Nürnberg (FAU), Glückstr. 11, 91054 Erlangen, Germany

**Keywords:** tissue engineering, dental pulp, stem cells, elasticity, scaffold, cellular mechanotransduction, regeneration

## Abstract

Dental pulp regeneration strategies frequently result in hard tissue formation and pulp obliteration. The aim of this study was to investigate whether dental pulp stem cells (DPSCs) can be directed toward soft tissue differentiation by extracellular elasticity. STRO-1-positive human dental pulp cells were magnetically enriched and cultured on substrates with elasticities of 1.5, 15, and 28 kPa. The morphology of DPSCs was assessed visually. Proteins relevant in mechanobiology ACTB, ITGB1, FAK, p-FAK, TALIN, VINCULIN, PAXILLIN, ERK 1/2, and p-ERK 1/2 were detected by immunofluorescence imaging. Transcription of the pulp marker genes BMP2, BMP4, MMP2, MMP3, MMP13, FN1, and IGF2 as well as the cytokines ANGPT1, VEGF, CCL2, TGFB1, IL2, ANG, and CSF1 was determined using qPCR. A low stiffness, i.e., 1.5 kPa, resulted in a soft tissue-like phenotype and gene expression, whereas DPSCs on 28 kPa substrates exhibited a differentiation signature resembling hard tissues with a low cytokine expression. Conversely, the highest cytokine expression was observed in cells cultured on intermediate elasticity, i.e., 15 kPa, substrates possibly allowing the cells to act as “trophic mediators”. Our observations highlight the impact of biophysical cues for DPSC fate and enable the design of scaffold materials for clinical pulp regeneration that prevent hard tissue formation.

## 1. Introduction

Multipotent stem cells (SCs) from human tissues are currently the most promising candidates for tissue regeneration and in vivo-like reconstruction of organoids and whole organs. While bone marrow-derived human mesenchymal SCs (hBMMSCs) have been intensely investigated in the context of soft tissue engineering [1,2,3], other SC loci within the human body have only been recently regarded as potential cell sources for likewise regenerative clinical therapies [4,5]. Among such SC niches, the dental pulp, i.e., the anatomical region within the pulp cavity of teeth, harbors a special population of SCs, called human dental pulp stem cells (in the following designated as DPSCs) [6]. DPSCs are derivates of the neural crest [7] and are therefore of ectodermal origin [8]. Of note, these cells also possess properties and surface protein markers characteristic of mesenchymal, neuronal, and embryonic SCs [5]. In particular, the great multilineage differentiation potential of DPSCs is reflected by the broad variety of different cell types, which are derivates of DPSCs and can be found in the dental pulp, including fibroblasts, peripheral neural cells (e.g., Schwann cells), endothelial and perivascular cells as well as odonto-osteoprogenitors [5,9]. Consequently, DPSCs offer great therapeutic potential in orofacial and cranial regenerative medicine.

Current standardized endodontic therapy for acute and chronic dental pulp disease involves the partial or total removal of inflamed or necrotic pulp tissue and the insertion of conventional endodontic filling materials [10]. Those, however, do not adequately mimic the in vivo situation of a healthy, vital pulp, which often results in therapeutic failure [11]. Therefore, innovative strategies that focus on regenerating or replacing dental pulp via tissue engineering are urgently needed [12,13,14]. However, pulp tissue engineering is complicated by the limited blood supply from the root apex [15] as well as the proximity of different tissues. They possess varying biochemical and biomechanical properties, thus creating a plethora of biological interfaces [16]. This is, e.g., reflected by the different extracellular matrix (ECM) composition as well as stiffnesses/elasticities of the hard and soft tissues within the dental pulp [17]. These facts lead to several as-yet unsolved problems in dental pulp regeneration: (i) the selection of the proper cell type for tissue engineering approaches, (ii) induction and in situ maintenance of cell differentiation and function, and (iii) the biochemical and biomechanical design of biomaterials aiming at optimally supporting dental pulp regeneration. Molecular characterization of DPSCs [18,19], including their developmental origin [20,21] and their multilineage differentiation potential [22] render this cell type an optimal candidate to address these shortcomings. Additionally, DPSCs are easily accessible [23], expandable [20], and ethically less questionable than other SCs [24].

Previous attempts to restore vital pulp tissue using DPSCs were hindered by the cells’ intrinsic disposition to build hard tissues, finally resulting in limited therapeutic success [12,13,25]. It is a widely accepted concept that DPSCs are prone to form tooth-like hard tissues [26,27,28], but little is known about the capacity of DPSCs to form soft pulp-like tissue. Additionally, it has recently been shown that DPSCs, like BMMSCs, can act as trophic mediators [13] by triggering nearby cells via chemotactic signals [29], i.e., for example, secreted transforming growth factor beta 1 (TGFb1), chemokine (C-C motif) ligand 2 (CCL-2) or colony-stimulating factor 1 (CSF1) [30] to regenerate surrounding tissues or to induce angiogenesis. Thus, the identification of factors, including biophysical cues that support and sustain soft tissue differentiation as well as a physiological secretory phenotype of DPSCs is of pivotal importance. It is known from hBMMSCs that they sensitively respond to extracellular matrix stiffness/elasticity, making defined cellular differentiation possible via biomechanical stimuli [31,32]. There is also the first scientific evidence that ECM stiffness influences DPSC behavior and differentiation [33]. However, evidence is lacking so far as to whether such factors also influence the cytokine expression of DPSCs. Offering a proper biomechanical environment can be achieved by the design of suitable biomaterials or scaffolds with the desired biophysical properties. Such materials, also described as cell-instructive [34], are thus of high importance for SC research, especially in the context of dental pulp regeneration [35].

On the molecular level, the transmission of extracellular, biomechanical information into the cell is a complex mechanism, also known as mechanotransduction. To date, several key players have been identified, which contribute to a fast and efficient exchange of biophysical information from the ECM across cellular membranes into the nucleus and back, i.e., outside-in and inside-out mechanotransduction [36]. Among them, focal adhesions (FAs) are prominent membrane multiprotein complexes, which are known to be involved in bidirectional mechanosignaling and cellular responses to biophysical cues, e.g., cell differentiation [37,38]. As parts of FAs, integrins are plasma membrane-embedded heterodimers with an extracellular domain [20,21], which bind ECM substrates such as fibronectin or collagen [39], as well as an intracellular domain that functions as a scaffold for many adapter proteins [24,40]. Focal adhesion kinase (FAK), a cytoplasmic tyrosine kinase, is one of these adapters proteins and works as a mechanotransduction switch [41,42]. In its activated, i.e., phosphorylated form, it transmits integrin-dependent signals downstream through phosphorylating other proteins [23]. One FAK target protein is paxillin [43]. Together with vinculin and talin, it indirectly links integrins to the actin cytoskeleton [44,45]. The latter is substantially influenced by integrin-dependent signaling and changes its composition and polymerization state in response to biomechanical stimuli [20]. To convert cytoskeletal tension into a cell adaptive response, i.e., a change in gene expression, the tension of the actin cytoskeleton must subsequently be transmitted into the nucleus. This is, amongst others, enabled by mitogen-activated protein kinases (MAPK) such as the extracellular signal-regulated kinase ERK 1/2 [46,47]. Finally, the change in gene expression can be monitored by characteristic expression profiles typical of e.g., dental pulp soft tissues.

Gene expression of the dental pulp reflects the wide diversity of pulp cell types. For example, matrix metallopeptidase 2 (MMP2), matrix metallopeptidase 3 (MMP3) [41], matrix metallopeptidase 13 (MMP13) [42], insulin-like growth factor 2 (IGF2) [43], and fibronectin 1 (FN) [44] are involved in tooth development, immune defense mechanisms, and dental pulp regeneration. Of interest, bone morphogenetic proteins (BMPs) contribute to the initial developmental interaction between the oral epithelium and the mesectoderm of the neural crest (mesenchymal-epithelial crosstalk). Bone morphogenetic protein 2 (BMP2) and bone morphogenetic protein 4 (BMP4) are associated with odontoblast differentiation, which can be found physiologically at the pulpal-dentine interface [45]. In the case of pulp irritation, these are the first cells to encounter bacteria or their respective toxins [46]. In particular, BMP4 has been shown to harbor anti-inflammatory properties [48]. The exact roles of BMPs, especially BMP2 in soft tissue regeneration, however, remain unclear [47].

These introductory remarks on DPSCs, the related gene expression profiles as well as mechanotransduction, indicate that these processes might be highly interconnected. Therefore, we hypothesized that DPSCs differentially sense and react to varying matrix elasticities. We aimed at investigating which cellular adaptations can be detected in response to the extracellular stiffness, and how DPSCs’ mechanotransduction is influenced by different growth substrates. Additionally, the corresponding transcriptional changes in established pulp markers genes, angiotrophic factors, as well as cytokines, were monitored. Our study provides insights into the mechanobiological responses and cellular adaptations of DPSCs to extracellular elasticity. The results are a first step in applying biomechanical-driven biomaterial design to tissue engineering and regeneration of dental pulp.

## 2. Materials and Methods

### 2.1. Isolation of DPSCs

All experiments were approved by the Committee of Ethics of the Medical Faculty of the Albert-Ludwigs-University Freiburg, Germany (vote number EK-153/15) and were carried out in accordance with the guidelines of the World Medical Association Declaration of Helsinki.

Dental pulp cells were obtained from non-carious human premolar teeth and third molars of three periodontal healthy patients (two female [both 14 years old] and one male [aged 20]) with written, informed consent. Briefly, extracted teeth were rinsed with povidone iodide (7.5%, Braun, Melsungen, Germany) and opened with a dentist’s drill at the level of the cementoenamel junction. Pulp tissue was extracted and cut into small pieces of 1–2 mm size. Cells were collected via explant culture in Alpha Minimum Essential Medium (MEM, Gibco™, Life Technologies, Carlsbad, CA, USA) with the following supplements: 10% fetal bovine serum (FBS, Sigma Aldrich, Munich, Germany), 1% GlutaMAX™ (Gibco™), and 1% Penicillin-Streptomycin (Sigma Aldrich). Cells were cultured under standard conditions (37 °C, 5% CO_2_) and medium was exchanged every 2–3 days (d). Cell passaging and culture maintenance was performed according to standard protocols. For this study, cells of passage (P) 1–2 were used and stored in liquid nitrogen until usage.

### 2.2. Magnetic-Activated Cell Sorting

Cells were magnetically sorted into STRO-1-positive and STRO-1-negative fractions using magnetic-activated cell sorting (MACS) microbead technology (Miltenyi Biotec GmbH, Bergisch Gladbach, Germany). For this purpose, cells were detached with Accutase^®^ solution (Sigma Aldrich) and subsequently incubated with human anti-STRO-1 antibodies (2 µg/106 cells, R&D Systems, Minneapolis, MN, USA) diluted in cold rinsing buffer (phosphate-buffered saline (PBS), Life Technologies, Darmstadt, Germany) with 2 mM EDTA (Sigma Aldrich) containing 0.5 % bovine serum albumin (BSA; Sigma Aldrich) for 30 min at 4 °C. After alternate washing steps, cells were incubated with 10 µL/10^6^ cells anti-mouse (ms) microbeads (Miltenyi Biotec GmbH) for 15 min at 4 °C. STRO-1-positive fractions were separated from the whole dental pulp cell collective in a magnetic field (MACs Multistand Miltenyi Biotec) using the MiniMACS™ separator (Miltenyi Biotec). Subsequently, STRO-1-positive cells were removed from the magnetic field and collected in 1 mL buffer using a plunger. Cells were incubated in supplemented MEM (Gibco^®^, supplements see above) at 37 °C and 5% CO_2_. The medium was exchanged every 2–3 days until cells reached a confluence of 80–90%. DPSCs were characterized as described previously [49].

### 2.3. Cell Culture on Elastic Substrates

Elastic polydimethylsiloxane (PDMS) substrates of 1.5 kPa, 15 kPa, and 28 kPa, respectively (ESS µ-dishes, ibidi GmbH, Munich, Germany), were coated with fibronectin solution (0.1%, Sigma-Aldrich) diluted 1:100 in phosphate-buffered saline (PBS, Life Technologies). Magnetically sorted STRO-1-positive cells were seeded at 1 × 10^4^/ 35 mm^2^ for indirect immunofluorescence (IIF, see below) analysis and 1 × 10^5^/ 35 mm^2^ and 1 × 10^4^/35 mm^2^ for quantitative PCR (qPCR) onto the different elastic substrates. Subsequently, they were cultivated for 3 days at 37 °C and 5% CO_2_. For IIF staining, cells were fixed using 4% paraformaldehyde (PFA; Sigma Aldrich) in PBS for 30 min at room temperature. For qPCR, cells were further processed with the help of the RNeasy Plus Micro Kit according to the manufacturer´s instructions (Qiagen, Hilden, Germany).

### 2.4. Indirect Immunofluorescence (IIF) Microscopy

For each target protein, cell donor, and elasticity, n = 3 samples were stained and examined via IIF after 3 days of incubation. IIF was performed according to standard procedures. In brief, blocking of unspecific binding sites and permeabilization was performed with 5% BSA (Stock solution 7.5 %, Sigma Aldrich) and 0.2% Triton X-100 (Sigma-Aldrich) in PBS for 1 h. Primary antibodies against ACTB (1:200, 8226 (ms)), ITGB1 (1:200, 52971 (rabbit [rb])), VINCULIN (1:200, 129002 (rb)) (all three from Abcam, Cambridge, UK), FAK (1:200, 3285S (rb)), pFAK (1:200, TYR397 8556S (rb)), TALIN-1 (1:200, 4021S (rb)), ERK1/2 (1:100, 4695S (rb)), pERK1/2 (1:250, 9101S (rb)) (all five from Cell Signaling Technology, Inc., Danvers, MA, USA), and PAXILLIN (1:100, 610620 (ms), BD Biosciences, Heidelberg, Germany) were diluted with 2 % BSA in PBS and incubated overnight at 4 °C. After repeated washing steps, secondary antibodies (1:200) were incubated with 2% BSA in PBS for 60 min at room temperature (Alexa 488 Goat Anti-Rabbit or Anti-Mouse IgG, respectively; Invitrogen AG, Carlsbad, CA, USA), followed by counterstaining with phalloidin™ 594 (1:40 in PBS; Invitrogen AG) for 40 min and DAPI 1:1.000 in PBS for 10 min (Sigma Aldrich). Samples were mounted using Fluoromount-G™ (Invitrogen AG) and photographed with an Olympus BX51 microscope (Shinjuku, Tokyo, Japan).

### 2.5. Quantitative (q) PCR

For RNA extraction, n = 6 samples were used from each donor and elastic substrate. For one donor, only three samples of each elastic substrate could be prepared due to low cell numbers. RNA from DPSCs was obtained using RNeasy Plus Micro Kit (Qiagen) according to the manufacturer’s protocol and stored at −80 °C. To verify the RNA integrity and quantity, samples were analyzed by the Experion automated electrophoresis system (Experion RNA HighSens Analysis Kit, Bio-Rad Laboratories, Munich, Germany) according to the manufacturer’s instructions. Subsequently, samples were amplified using the RT2 PreAMP cDNA Synthesis Kit (Qiagen) according to the manufacturer’s protocol. Samples were adjusted to the equal RNA concentrations of 8 ng/µL. cDNA samples were quantitatively amplified in duplicates to evaluate the expression of the following genes: angiopoietin 1 (ANGPT1), bone morphogenetic protein 2 (BMP2), bone morphogenetic protein 4 (BMP4), matrix metallopeptidase 2 (MMP2), matrix metallopeptidase 3 (MMP3), matrix metallopeptidase 13 (MMP13), fibronectin 1 (FN1), insulin-like growth factor 2 (IGF2), vascular endothelial growth factor (VEGF), chemokine (C-C motif) ligand 2 (CCL2), transforming growth factor beta 1 (TGFB1), interleukin 2 (IL2), angiogenin (ANG), colony-stimulating factor 1 (CSF1), beta-actin (ACTB), and hypoxanthine phosphoribosyltransferase 1 (HPRT1) by using pre-validated gene-specific RT2 qPCR primer assays (Qiagen). Real-time qPCR was performed using 1 µL of each cDNA sample in duplicates in a 25 µL reaction volume containing RT2 SYBR^®^ Green Mastermix (Qiagen) according to the manufacturer’s instructions. Samples were pipetted into a 96-well-plate (Bio-Rad Laboratories), sealed (Microseal^®^, Bio-Rad Laboratories), and centrifuged (Centrifuge 5430R, Eppendorf AG, Hamburg, Germany) for 1 min at 1000× *g*. Samples were incubated at 95 °C for 10 min, then at 95 °C for 15 s, and at 60 °C for 60 s for 40 cycles. For each run, negative reverse transcription and negative template controls were included. Data were collected by CFX96 Manager Software Version 1.0 (Bio-Rad). Samples were checked for Ct-value consistency (ΔCt < 0.5). Relative quantities of the respective genes of interest were normalized to the relative quantities of HPRT1 as a housekeeping gene. Results were analyzed and plotted using the RT² Profiler PCR Data Analysis Software (Qiagen) at http://www.qiagen.com/geneglobe (accessed on 9 July 2020) representing fold-change results. For true-to-life analyzation, instead of using an unphysiological polystyrol control, gene expression of DPSCs grown on 1.5 kPa (test group 1) and 28 kPa (test group 2) substrates were compared to those on 15 kPa (control group) elasticities.

### 2.6. Statistics

For each condition, paired *t*-tests were used to evaluate the influence for qPCR results, and the resulting *p*-values were adjusted by applying the Bonferroni method to correct for multiple testing. Results were considered statistically significant if *p* < 0.05. The calculations were performed with the statistical software STATA 17.0 (StataCorp LLC., College Station, TX, USA).

## 3. Results

DPSCs adapt their morphology in response to the surrounding substrate elasticity. To analyze the cell morphology of DPSCs on substrates with different elasticities, cells were seeded on commercially available polydimethylsiloxane (PDMS) culture dishes with a stiffness of 1.5 kPa, 15 kPa, or 28 kPa, respectively. Subsequently, cells were grown under standard cell culture conditions (37 °C, 5% CO_2_) for 1, 2, or 3 days to determine early morphological adaptations of the DPSCs to the surrounding mechanical environment. Morphological differences of DPSCs growing on the different substrates could be observed already after one day. On 1.5 kPa (in the following designated as “soft”) substrates, DPSCs exhibited a round, polygonal shape with only small cell branches in all directions (Figure 1a,d,g). At 15 kPa (in the following designated as “intermediate”) elasticity, cells appeared in a star-like shape featuring intermediate branching (Figure 1b,e,h). In comparison, on PDMS with an elasticity of 28 kPa (in the following designated as “stiff”), DPSCs had rather elongated or spindle-shaped cell bodies with a substantial number of branches (Figure 1c,f,i). As can be seen from the increasing cell densities in the course of the three days (Figure 1a vs. Figure 1g, Figure 1b vs. Figure 1h, and Figure 1c vs. Figure 1i; all same magnification), DPSCs can adhere and proliferate on all three substrates. In summary, DPSCs adapt their cell shape in response to environmental elasticity. Of interest, an increasing substrate stiffness leads to the elongation and branching of the cells.

The expression, activation, and subcellular localization of mechanosensing and mechanotransducing proteins are differentially regulated by environmental stiffness in DPSCs

Since cell morphology is strongly connected to cellular adhesion and the cytoskeleton [50,51], components of these molecular networks were investigated next. The actin cytoskeleton is composed of alpha (α)- and beta (β)-actin and a plethora of actin-organizing proteins. The actin fibers render the cell resistant to mechanical stress and are organized in response to mechanical stimuli such as extracellular elasticity. To see if the above-described cellular morphologies are related to actin fiber organization, beta-actin was analyzed via indirect immunofluorescence (IIF) staining. The overall cell morphology differed between the soft (Figure 2a), intermediate (Figure 2b), and stiff (Figure 2c) substrate upon IIF examination, mirroring the light microscopic findings from Figure 1. In particular, the DPSCs revealed obvious differences in actin filament organization after fixation on day three of incubation. Round-shaped DPSCs cultivated on soft substrates exhibited randomly distributed filaments (Figure 2a,d). This is in accordance with the flat and short, multi-directional cell branches. Contrary to this, filaments of the more elongated cells grown on intermediate and stiff substrates seemed more organized and aligned in a pattern parallel to the longest cellular axis (Figure 2b,c,e,f). Beta-actin was strongly and densely expressed especially in cells on stiff substrates, with single fibers being hardly distinguishable (Figure 2c,f). Thus, the beta-actin organization pattern is also responsive to the stiffness of the PDMS substrate and is strongly related to the microscopically visible cellular morphology.

Actin cytoskeletal rearrangement in response to mechanical stimuli is, however, just a common final path of many cellular signaling processes, which convert extracellular biophysical cues into intracellular biochemical responses [40]. The latter process is also called mechanotransduction and involves many proteins, including the structural components of focal adhesions (FAs). FAs are in contact with the extracellular matrix (ECM) and the actin cytoskeleton. Among the FA components, integrins possess an extracellular domain, which interacts with the ECM (mechanosensing), and an intracellular domain, which stimulates downstream mechanosignaling events (mechanotransduction) [43]. Focal adhesion kinase (FAK) is a cytosolic tyrosine kinase associated with integrins and functions as a mechanotransduction switch [42]. Its phosphorylated form, p-FAKY397 (phosphorylated tyrosine residue at position 397) is an indicator of activated mechanosignaling. Therefore, the protein abundance and cellular distribution patterns of beta-1-(β1) Integrin, FAK, and pFAKY397 were analyzed via IIF after three days of incubation.

Integrin β1 was present in the short extensions of the cells cultivated on 1.5 kPa substrates (Figure 3a). In DPSCs cultivated on 15 kPa and 28 kPa PDMS dishes, the integrin signal was rather evenly distributed in the cell body (Figure 3b,c). Upon cultivation on 28 kPa substrates, integrin staining intensity was the highest when compared to 1.5 kPa (intermediate intensity) and 15 kPa (lowest intensity) (Figure 3c compared to Figure 3a,b).

Regarding staining for total FAK, i.e., non-phosphorylated and phosphorylated FAK, cells cultured on soft substrates showed a diffuse distribution of the protein with a slight tendency towards a perinuclear staining pattern (Figure 3d). A similar FAK localization was observed in cells cultivated on intermediate substrates, and the staining signal appeared more intense in this group (Figure 3e). In contrast, on stiff substrates, the cells stained less intensely for FAK, and the protein was detected rather in the elongated cell branches than beneath the nucleus when compared to the other experimental groups (Figure 3f, magnification).

Subsequent analysis of p-FAKY397 revealed a comparable result for 1.5 kPa and 15 kPa substrates. In both setups, the phosphorylated and hence active isoform of FAK was found closer to the cell periphery compared to the respective total FAK staining (Figure 3g,h). This pattern was especially emphasized on the soft substrates (Figure 3g). On the stiff substrates, p-FAKY397 was also mainly detected in the cell periphery (Figure 3f,i).

In summary, the experiments prove (i) the presence of the respective proteins in this cell type and (ii) some differences in stiffness-dependent subcellular localizations.

Since integrins, FAK, and p-FAK are no direct binding partners of actin filaments, we were interested in whether mechanotransducing proteins that are more intimately involved in coupling integrin/FAK signaling with actin cytoskeletal remodeling, might be influenced by the different substrate elasticities. Therefore, talin, vinculin, and paxillin, which are all FA constituents and directly or indirectly bind and/or modulate the actin cytoskeleton, were examined via IIF after 3 days of incubation. Immunofluorescence signals for all three proteins were present in all DPSCs (Figure 4). In DPSCs cultured on soft substrates, talin was detected in close association with the nucleus (Figure 4a). A comparable signal intensity could be observed in DPSCs on intermediate elasticities, while the protein was more evenly distributed within the cell body (Figure 4b). With regards to stiff substrates, talin was uniformly expressed throughout the cell body (Figure 4c).

Cells on 1.5 kPa substrates showed a similar expression pattern for vinculin as for talin surrounding the cell nucleus (Figure 4d). DPSCs on intermediate substrates were found to stain intensely for vinculin close to their nucleus (Figure 4e), while cells on stiff scaffolds expressed a strong signal for vinculin in the long cell extensions (Figure 4f).

Paxillin expression could be observed in a diffuse pattern close to the cell nucleus in DPSCs on soft and intermediate substrates (Figure 4g,h). Elastic substrates with a stiffness of 28 kPa harbored DPSCs, which stained intensely for paxillin throughout the entire cell body, but in contrast to softer substrates, the protein was also found in cell extensions (Figure 4i). Taken together, the three proteins associated with cytoplasmic mechanotransduction and cytoskeletal organization, i.e., talin, vinculin, and paxillin, were predominantly localized in the perinuclear region of DPSCs on soft and intermediate PDMS substrates, whereas a marked enrichment in cellular extension could be detected in cells grown on stiff matrices.

In addition to cytoskeletal remodeling, mechanotransduction pathways interact with classical cellular signaling cascades, which direct cellular behavior. The extracellular signal-regulated kinases 1/2 (ERK 1/2) are part of the mitogen-activated protein kinase (MAPK) pathway. Its activated isoform, phosphorylated (p) ERK 1/2, therefore is an indicator of active mechanobiological signaling. For gaining insights into the mechanistic pathways underlying the cellular response to different environmental elasticities, DPSCs were stained for total ERK 1/2 (Figure 5a–c) and its activated isoform p-ERK 1/2 (Figure 5d–f) via IIF after fixation on day 3 of incubation. ERK 1/2 was detected in all samples and was diffusely distributed within DPSCs growing on soft substrates (Figure 5a). A higher matrix stiffness led to a concentration of the protein in the perinuclear region (Figure 5b,c). Of note, cells from all groups showed an intense signal for the activated/phosphorylated ERK 1/2 kinases in all samples, especially on stiff substrates, which exhibited a pan-cellular protein distribution (Figure 5f). Thus, MAPK signaling cascades are efficiently activated by the different extracellular environments with no clear distinction between the respective elasticities.

Soft substrates enhance the transcription of pulp soft tissue marker genes and intermediate elasticity substrates support angiotrophic cytokine gene expression in DPSCs

The IIF analysis of mechanobiologically relevant proteins led to the first insights that the varying extracellular substrates differentially affect the protein abundance and subcellular localization of these mechanotransduction key players. Since mechanotransduction affects cellular differentiation, we hypothesized that the distinct protein patterns are related to different gene expression signatures associated with the multilineage potential of DPSCs. The transcription of genes characteristic for the differentiation of DPSCs into dental pulp soft tissue cells was measured as a function of the elasticities via RT-qPCR after 3 days of culture. A short incubation time was selected to show immediate cell responses and sensitivity to biomechanical environmental changes, as the cell fate is decided within a relatively short time frame [52].

Additionally, cytokine genes were selected as a means of estimating the cells’ ability to act as angiotrophic mediators. Since the different substrate elasticities were chosen to represent typical elastic properties as found in the dental pulp in vivo, a conventional polystyrene culture dish control was not included because it possesses elastic moduli of between 3–3.5 GPa [53], which is far beyond the physiological range [27]. Therefore, DPSC gene expression in the 1.5 kPa (“soft”) and 28 kPa (“stiff”) groups were statistically compared to cells cultured on 15 kPa (“intermediate”) substrates.

Generally, DPSCs cultivated on soft substrates showed the highest transcription of pulp soft tissue marker genes among all groups (Figure 6a). This can be seen from the proportionally high mRNA expression of bone morphogenetic protein 2 (BMP2), matrix metallopeptidase 2 (MMP2), matrix metallopeptidase 3 (MMP3), fibronectin 1 (FN1), and insulin-like growth factor 2 (IGF2) when compared to the intermediate elasticity substrate. Matrix metallopeptidase 13 (MMP13) and bone morphogenetic protein 4 (BMP4) were the only genes that showed a decreased transcription in DPSCs cultured on 1.5 kPa substrates in comparison with an elasticity of 15 kPa. For BMP2, MMP2, and FN1, gene transcription was similar between stiff and intermediate substrates but opposite to the soft substrate. Of interest, mRNA expression of IGF2 was most pronounced on 28 kPa substrates. Cells on the latter substrate exhibited MMP3 and MMP13 gene levels comparable to soft substrates (Figure 6a). No statistically significant differences in gene transcription were detected upon the comparison of soft substrates vs. intermediate substrates or stiff substrates vs. intermediate substrates.

Regarding cytokine gene transcription (Figure 6b), angiogenin (ANG), vascular endothelial growth factor (VEGF), transforming growth factor beta 1 (TGFB1), and interleukin 2 (IL2) showed a reduced number of transcripts on soft substrates when compared to the intermediate substrates. This applies to the same transcripts when comparing stiff with intermediate scaffolds. Angiopoietin 1 (ANGPT1), chemokine (C-C motif) ligand 2 (CCL2), and colony-stimulating factor 1 (CSF1) were pronounced in similar amounts in 1.5 kPa- and 15 kPa-grown DPSCs, whereas these transcripts exhibited a low expression profile on stiff substrates. No statistically significant differences in gene transcription were detected upon the comparison of soft substrates vs. intermediate substrates or stiff substrates vs. intermediate substrates. In summary, DPSCs cultivated on 15 kPa substrates appeared to provide the strongest mRNA expression of cytokine genes related to angiogenesis and cell interactive regulations (Figure 6b), while soft tissue marker gene transcription was most pronounced in DPSCs cultivated on soft substrates (Figure 6a).

## 4. Discussion

Dental pulp regeneration is a complex undertaking, which is currently limited by anatomical and cell biological, and material restraints. Although DPSCs are a promising cell population for this purpose, many facets of their biology as well as their interaction with biomaterials are so far incompletely understood. We, therefore, aimed at characterizing DPSCs behavior on fibronectin-coated PDMS substrates with different elasticities to analyze whether the varying biomechanical stimuli influence DPSC morphogenesis and differentiation. To the best of our knowledge, this is the first report showing that extracellular elasticity alone can direct DPSCs to adopt a soft tissue specification, i.e., a typical round morphology in combination with a characteristic gene expression profile. The only existing study on biomechanical cues on DPSCs so far showed that stiff matrices together with inorganic additives are required for the DPSCs’ capacity of hard tissue formation [27]. Beyond that, here we show that extracellular elasticity impacts the expression and distribution of mechano-sensing and -transducing proteins as well as the transcription of soft tissue and cytokine-encoding genes. Taken together, we successfully showed that DPSCs, as reported for BMMSCs [31], react very sensitively to surrounding physical attributes. Additionally, our observations indicate that the biophysical properties of scaffold biomaterials can be designed to enable DPSCs differentiation towards efficient soft pulp tissue regeneration.

In this study, pulp cells were enriched via STRO-1-dependent magnetic-activated cell sorting. STRO-1+ dental pulp cells have been shown to exhibit excellent SC properties, including a high proliferative capacity, multilineage differentiation potential, and a low tendency toward cellular senescence [49]. Standardization of DPSCs isolation and enrichment protocols is a central precondition for reproducible and safe application of this cell type in regenerative treatments in the clinic. Therefore, future characterization of DPSCs interaction with different extracellular substrates and biomaterials should be performed with comparable and defined cell populations, i.e., purified dental pulp cells according to reasonable experimental protocols.

The first step in analyzing the DPSCs’ behavior on the different substrates was the light microscopic evaluation of cellular morphology. After only 1 day of incubation, DPSCs clearly exhibited different morphologies. Of note, longer culture periods favor the adoption of similar cell shapes in DPSCs [27]. This highlights the urgency of investigating very early/immediate effects of biophysical cues, i.e., analyzing the cells in an early phase of culture [52]. There is compelling evidence that the round cell shape with very little cell branching as observed on the 1.5 kPa substrates is associated with a lineage specification toward soft tissue [31,54]. Similar experimental findings have been reported for mesenchymal SCs (MSCs), where round cell shapes are associated with adipogenic (soft tissue) differentiation [40,55,56]. Cells at 28 kPa elasticity developed an elongated cell shape with long cell extensions, which resembles the phenotype of MSCs and SCs grown on stiff substrates [56,57,58,59]. These morphologic findings suggest a link between hard tissue specification of DPSCs, i.e., osteo- [54] or dentino-genic differentiation [60,61] as triggered by stiff substrates. DPSCs grown on 15 kPa substrates exhibited a star-shaped phenotype, which shows phenotypic similarity to hBMMSCs with neuron-like differentiation [31]. These first experiments have clearly demonstrated the impact of extracellular stiffness on DPSCs morphology and pose the question of to what extent this influences intracellular signaling pathways.

Therefore, the actin cytoskeleton was investigated next since it is one main factor determining cellular morphology. In particular, actin fibers are well-established target structures of many mechanobiologically-relevant signaling pathways that might be differentially regulated by varying extracellular elasticities. Filament alignment of beta-actin was shown to be loose in cells cultivated on soft substrates, substantiating the interpretation of a soft tissue specification [31,54]. In comparison, actin fibers on intermediate and stiff substrates appeared more organized and closely aligned parallel to the longest cellular axis. Fiber alignment as well as the density of actin filaments is associated with the tension of the actin cytoskeleton [31,54,55,62]. This suggests that especially the cytoskeleton of DPSCs seeded on the stiff 28 kPa substrates possess a considerable tension. This was also observed during hard tissue lineage specification of hBMMSCs, where the cells have to resist the external forces of the respective substrate [31]. This again points in the direction that there is a correlation between the hard tissue specification of DPSCs and a high extracellular stiffness [54,63], further substantiating the morphological findings. Recent studies have characterized the in vivo-like viscoelastic and mechanic properties of dental pulp [64] and determined an elastic modulus of 5.5 kPa for dental pulp [65]. The viscoelasticity of dental pulp, i.e., the scaffold material, is crucial for the function of the individual cell types [60] for pulp regeneration as well as maintaining homeostasis.

The morphological and cytoskeletal findings indicated that mechanosignaling pathways are directly affected by the different substrate stiffnesses. Therefore, FAs components, i.e., β-integrin, FAK, p-FAKY397, talin, vinculin, and paxillin were analyzed next. In general, the most intense staining signals could be detected in DPSCs cultivated on stiff substrates. This means that for mostly integrin-ß1, vinculin, and paxillin were found to be expressed more intensely and localized at sites of focal adhesions in cells on stiff substrates. This observation is in line with corresponding findings for hBMMSC [31]. This supports the notion that hard tissue formation might be favored in DPSCs cultured on stiff substrates [54,63]. A defined localization of FA components near cellular margins is also an indicator of strongly activated mechanosignaling [66,67,68]. The remarkable p-ERK 1/2 signal in this experimental group is in accordance with these findings. Diffuse and perinuclear localization of talin, vinculin, or FAK in DPSCs grown on soft and intermediate substrates contributes to the body of evidence that these cells are prone to a soft tissue differentiation and that FA signaling and thus FAK activation and cytoskeletal tension are comparatively low [67,68]. Taken together, DPSCs exhibit hBMMSC-like behavior in response to substrate elasticity. Furthermore, stiff substrates led to an intensification of ERK 1/2 activation, indicating the involvement of ERK 1/2 in DPSC mechanotransduction, being consistent with data on the hard tissue, i.e., osteogenic, differentiation in MSCs [54,55]. Conversely, a lower extracellular stiffness is permissive for morphological and mechanosignaling features of soft tissue differentiation in DPSCs.

Since changes in morphology and mechanotransduction are strongly connected to lineage specification [69] and have previously been proven to be directed by microenvironmental elasticity [31], we assessed the mRNA expression levels of selected pulp marker and cytokine genes. We wanted to simulate physiological conditions, and therefore a cell culture polystyrol dish control was omitted for the experiments due to elasticities in the GPa range. The lack of statistical significance in the gene transcription experiments might be either related to sample pooling in the study, which was selected in order to prevent bias in differentiation by cell density [38] or to the absence of a plastic control, or to the short incubation time compared to studies using BMMSCs [27,31,70]. Nevertheless, soft substrates with 1.5 kPa elasticity clearly led to the transcription of the expected soft tissue genes in DPSCs. This observation is in line with the interpretation that stiffer matrices foster a hard tissue differentiation in DPSCs similar to BMMSCs [31]. Interestingly, the mRNA expression of MMP13 as a key pulp tissue marker [71] was highest in the 15 kPa elasticity group. This could be interpreted as a disposition of DPSCs growing on substrates with an intermediate stiffness to prevent hard tissue differentiation [72]. The increased transcription of MMP2 on soft and intermediate substrates (1.5 kPa and 15 kPa) indicates the involvement of MMP2 in matrix turnover [30]. MMP3 displayed higher mRNA expression levels both on 1.5 kPa and 28 kPa substrates, which could positively stimulate wound healing [71] as well as the pulp regeneration capacity in these DPSCs [73]. The higher transcription of MMP3 could further be connected to a DPSC phenotype, which is involved in tertiary dentine formation [74] or stimulates angiogenesis [71]. Taken together, the transcript expression levels of different MMP isoforms are thus strongly connected to microenvironmental elasticity and in turn are likely to influence the ECM composition in vivo, generating a feedback loop.

Bone morphogenetic protein 2 (BMP2), which is normally associated with stimulating hard tissue formation in MSCs [75], was found with increased transcription on soft substrates in this study. This finding supports recent claims that a potential osteogenic effect of BMP2 may be more dependent on both cell type and environmental cues [76]. Additionally, a recent study by Aksel and Huang [77] showed that incubation with BMP2 in addition to osteo-/odontogenic medium had little to no impact on the mineralization of DPSCs. Furthermore, bone morphogenetic proteins have been reported to participate in early dental differentiation decisions [45], and thus may play their part in pulp regeneration. These observations highlight the urge for investigating the precise role of BMP2 in the dental pulp and show that lineage differentiation in DPSCs may not necessarily resemble BMMSCs. Of interest, BMP4 was least expressed in DPSCs on soft substrates and showed a stronger mRNA expression on stiffer substrates (15 kPa and 28 kPa). These results are in line with an osteogenic/odontogenic differentiation as promoted by BMP4 [78]. Consequently, hard tissue formation of DPSCs is promoted by rather stiff substrates [31,54].

The cytokine gene transcription was most pronounced in DPSCs cultivated on intermediate substrates with 15 kPa. In particular, VEGF, ANG, and CSF1, i.e., angiogenetic factors [79], as well as immune response and homeostasis regulating factors TGFb1 [80,81] and IL-2 [82,83], could be detected in the intermediate group. TGFb1 is a key mediator for the maintenance of mechanical loading of the pulp and is responsible, among other mechanisms, for periostin expression in PDL fibroblasts and pre-odondoblasts. An increased expression of periostin in DPSCs can in turn be triggered via biomechanical stimulation. Interestingly, the downregulation of periostin in DPSCs is associated with a higher risk of pulp obliteration [84].

Increased mRNA expression of CCL-2, an anti-inflammatory factor responsible for initial immune cell recruitment [48], angiogenesis, and homeostasis [85] however, could be detected on the soft and intermediate substrates rather than the stiff ones. Overall, it is tempting to speculate that DPSCs serve as trophic mediators in vivo. This would mean that these SCs are not only a pool for cell regeneration but that their secretory function is needed to sustain the dental pulp and to enable tissue repair [86]. This hypothesis must be supported by further research in the future.

In summary, lineage specification of DPSCs can be modulated towards soft tissue formation by attachment to soft substrates. This is in line with corresponding findings from hBMMSCs [31]. Of interest, intermediate substrates promoted cytokine gene transcription and therefore may enable DPSCs to act as trophic mediators and pro-angiogenic factors in the homeostasis of the dental pulp [13]. DPSCs on stiff substrates are prone to form hard tissues [32,54], which might be interesting in terms of periodontal regeneration, while it needs to be prevented for pulp regeneration [27,87].

## 5. Conclusions

When DPSCs encounter hard tissue substrates in an empty pulp cavity following endodontic treatment, the cells are supposed to differentiate and form hard tissues. To prevent the obturation of the pulp cavity, scaffold materials with suitable biophysical features enabling a defined differentiation of DPSCs thus appear indispensable. According to our data, an elasticity of 1.5 kPa seems to be particularly suitable for pulp soft tissue regeneration, while 15 kPa could stimulate the attraction of surrounding cells and support dental pulp homeostasis. This elasticity range reflects the elastic modulus of 5.5 kPa of the natural pulp tissue [65].

The use of DPSCs for soft tissue regeneration in the pulp cavity could represent a tremendous advance in endodontic therapy. Thus, the results of our study contribute to the body of evidence that biomechanically driven material design is useful in targeted cell differentiation. This knowledge will be applied in the development of innovative endodontic tissue engineering materials and therapies.

## Figures and Tables

**Figure 1 bioengineering-10-00323-f001:**
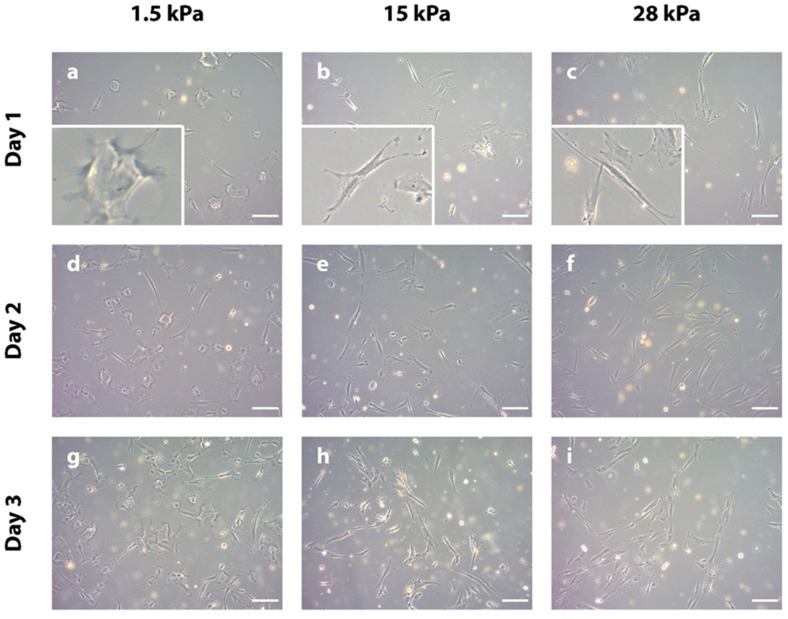
Representative light microscopic images of DPSCs grown on polydimethylsiloxane (PDMS) substrates with different elasticities (1.5 kPa, 15 kPa, and 28 kPa, respectively, for 1, 2, and 3 days each). (**a**,**d**,**g**) On 1.5 kPa substrates, DPSCs show round, polygonal cell shapes with only short branching as indicated by the inset in (**a**). (**b**,**e**,**h**) A higher substrate stiffness of 15 kPa leads to a different cell phenotype, i.e., star-like shaped DPSCs with intermediate branching as represented by the inset in picture (**b**). (**c**,**f**,**i**) Of interest, the highest substrate stiffness, i.e., 28 kPa, leads to a spindle-shaped morphology of DPSCs with long extensions (see also inset in picture (**c**)). Scale bars represent 50 µm.

**Figure 2 bioengineering-10-00323-f002:**
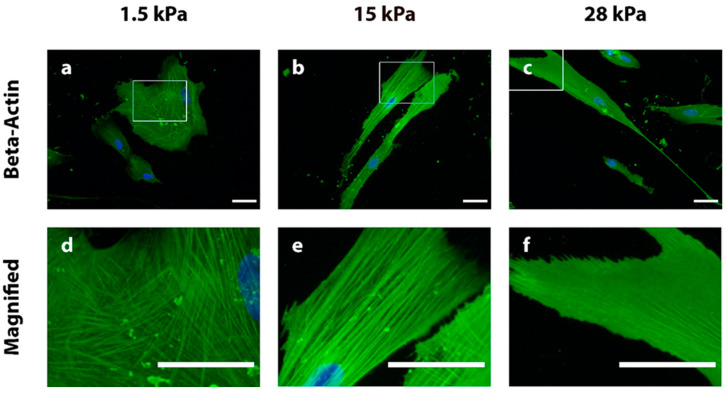
Indirect immunofluorescence (IIF) staining of beta-actin (green) and the nucleus (DAPI staining, blue) in DPSCs grown on PDMS substrates with elasticities of 1.5 kPa (**a**,**d**), 15 kPa (**b**,**e**), and 28 kPa (**c**,**f**) on day three. Lower line d–f: magnifications show beta-actin filament alignment in more detail. Round-shaped DPSCs with short cell extensions on 1.5 kPa substrates (**a**,**d**) show a diffuse, multi-directional, and loose actin filament alignment. Contrary to this, cells on the intermediate (**b**,**e**) and stiff (**c**,**f**) substrate present with a high density of actin fibers, which are aligned parallel to the longest cellular axis. Details are given in the main text. Scale bars represent 50 µm.

**Figure 3 bioengineering-10-00323-f003:**
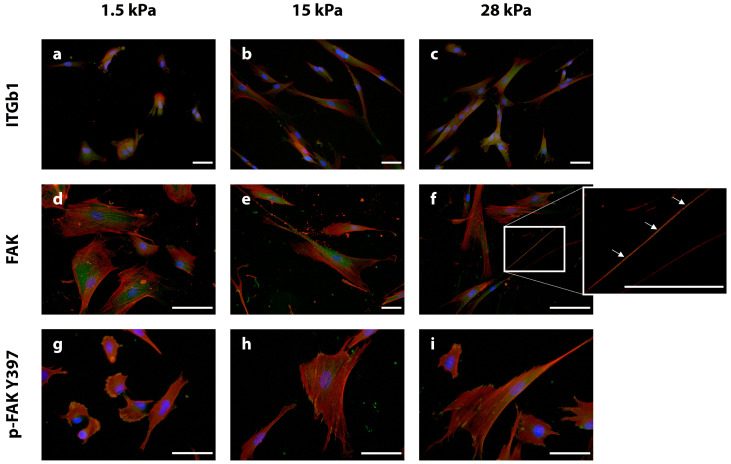
Immunostaining of the focal adhesion (FA) proteins integrin beta 1 (ITGb1) (**a**–**c**), focal adhesion kinase (FAK) (**d**–**f**), and phosphorylated (p) FAK at position Y397 (p-FAK) (**g**–**i**) (protein of interest stained in green, respectively). The cytoskeleton was stained with phalloidin (red) and the cell nuclei with DAPI (blue). On the soft (1.5 kPa) substrates, ITGb1 was found diffusely distributed throughout the cell body, but especially in the short, thick cell extensions (**a**). On the intermediate (15 kPa) substrates, ITGb1 appeared evenly distributed within the cells (**b**). The strongest immunofluorescence signal for ITGb1 could be detected on the stiff (28 kPa) substrates, and the protein was distributed evenly throughout the entire cell-covered area (**c**). FAK could be found close to the nucleus in DPSCs on soft substrates (**d**). Like DPSCs on soft scaffolds, FAK was found close to the nucleus in intermediate substrates but was also localized in the cell periphery (**e**). In contrast to softer substrates (**d**,**e**), FAK on stiff substrates was found within the long cell extensions ((**f**), see also higher magnification, white arrows). Staining for active FAK (p-FAK) on 1.5 kPa substrates showed high concentrations at cell margins (**g**). In DPSCs grown on intermediate and stiff substrates, p-FAK was found to be diffusely distributed within the cell body. The overall staining intensity was lower for both 15 kPa and 28 kPa in comparison with 1.5 kPa ((**h**,**i**) vs. (**g**)). The scale bars represent 50 µm.

**Figure 4 bioengineering-10-00323-f004:**
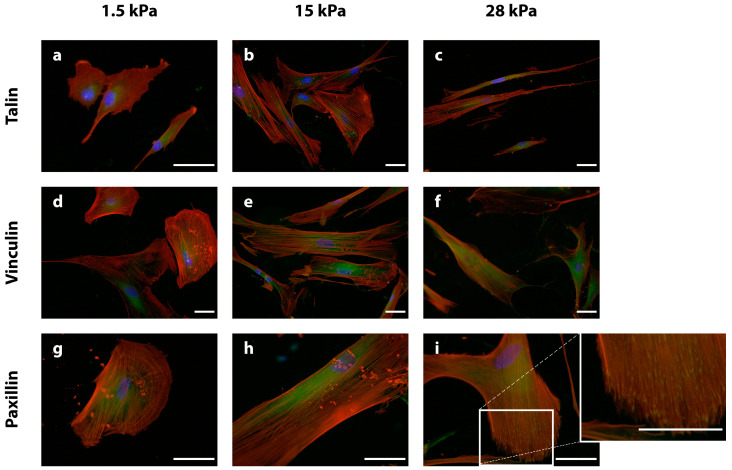
Immunostaining of the focal adhesion (FA) adapter proteins talin (**a**–**c**), vinculin (**d**–**f**), and paxillin (**g**–**i**) (the respective proteins of interest are depicted in green) after fixation on day three. The cytoskeleton is stained with phalloidin (red), and the nuclei are stained with DAPI (blue). In general, the immunofluorescence signal for DPSCs on 1.5 kPa substrates (**a**,**d**,**g**) was found near the nucleus for all adapter proteins (talin, vinculin, and paxillin), whereas the signal shifted toward the cell margins with increasing stiffness (15 kPa (**b**,**e**,**h**), 28 kPa (**c**,**f**,**i**)). Staining of paxillin in DPSCs grown on 28 kPa matrices revealed a clearly visible concentration of the protein at the cell margins ((**i**), see also the higher magnification). The scale bars represent 50 µm.

**Figure 5 bioengineering-10-00323-f005:**
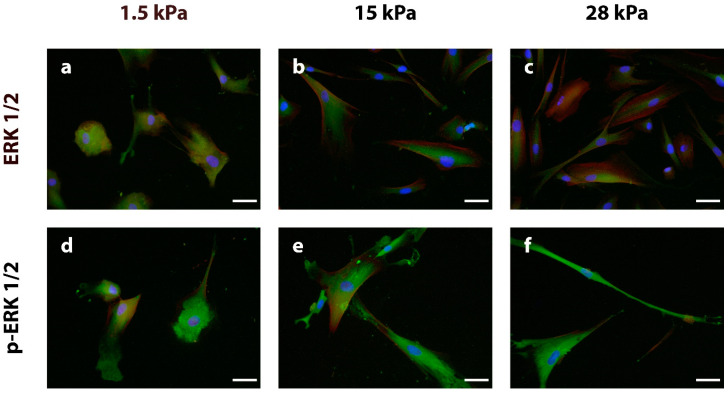
Immunostaining of the mitogen-activated protein kinase (MAPK) signal transmission protein extracellular signal-regulated kinase ERK 1/2 (**a**–**c**) and its activated, phosphorylated isoform p-ERK 1/2 (**d**–**f**) after fixation on day three (the respective proteins of interest are stained in green). The cytoskeleton is stained with phalloidin (red), while the nuclei are stained with DAPI (blue). ERK 1/2 can be found in the whole cytoplasm of DPSCs growing on soft substrates (**a**), whereas it is concentrated in the nuclear region of intermediate (**b**) and stiff (**c**) matrices. A separate analysis of activated p-ERK 1/2 revealed intense staining in DPSCs on all three substrates (**d**–**f**) with a strong pan-cellular expression on stiff substrates (**f**). The scale bars represent 50 µm.

**Figure 6 bioengineering-10-00323-f006:**
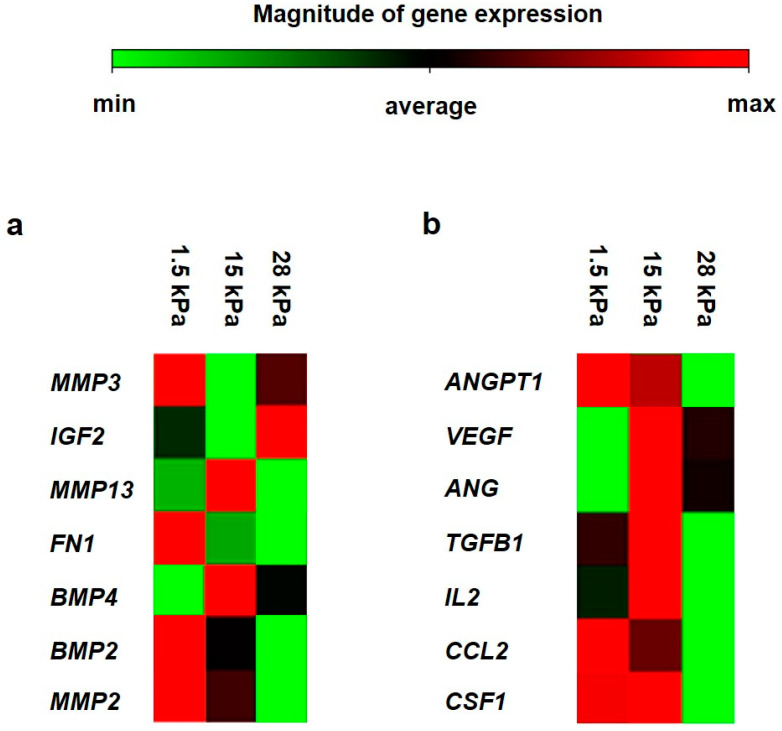
Relative gene transcription levels of selected soft tissue markers and cytokines on day 3 of cultivation. Columns of the clustergram: elasticities (from left to right: 1.5 kPa (soft), 15 kPa (intermediate), 28 kPa (stiff), rows of the clustergram: genes investigated: (**a**): pulp marker genes: matrix metallopeptidase 13 (MMP13), insulin-like growth factor 2 (IGF2), bone morphogenetic protein 4 (BMP4), bone morphogenetic protein 2 (BMP2), matrix metallopeptidase 2 (MMP2), matrix metallopeptidase 3 (MMP3), fibronectin 1 (FN1), (**b**): angiotrophic cytokine genes: angiogenin (ANG), vascular endothelial growth factor (VEGF), angiopoietin 1 (ANGPT1), chemokine (C-C motif) ligand 2 (CCL2), colony-stimulating factor 1 (CSF1) transforming growth factor beta 1 (TGFB1), interleukin 2 (IL2). The relative gene transcription in comparison with a housekeeping gene (see Materials and Methods) is represented as colors (see the color scale at the top of the figure).

## Data Availability

Stem cell characterization data referred to in this study are available in [Cells 2022, 11(20), 3204; https://doi.org/10.3390/cells11203204].

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
