# Peer review of "Biomechanical Modulation of Dental Pulp Stem Cell (DPSC) Properties for Soft Tissue Engineering"

_bioengineering, 2023, doi:10.3390/bioengineering10030323_

Round 1

Reviewer 1 Report

In their study, the authors report on the effect of substrate stiffness on cell morphology, cytoskeleton structure, transcription pattern of cytoskeleton regulatory proteins, as well as some cytokines and dental pulp markers. All these parameters are influenced by matrix stiffness. The data are interesting and innovative for both basic stem cells research  and potential application in regenerative dentistry.

However,  I have some questions concerning the details of the study

11.  ACTB and HPRT1 were used in the qPCR experiments as reference genes. However, as shown in Figure 2, ACTB might be a gene of interest in this study – its activity might change during mechanical stimulation. As shown by Liu et al (https://doi.org/10.1002/stem.160) ACTB is better to be avoided as a reference gene in in mechanobiology studies.  Nazari et al (10.1186/s40781-015-0050-8) and Zucherato et al (https://doi.org/10.1038/s41598-021-84884-5) have reported ACTB as a reference gene not suitable for cells of mesenchymal stem cell phenotype (DPSC are ectomesenchymal but they do have a similar phenotype).

22. The results in Figure 3 (especially in middle and bottom rows) seem not very convincing for me. I would recommend to add quantified results or provide better images in support of authors’ claims.

Some minor points:

11.  L.193  ‘BACT’ -  is it a typo instead of ‘ACTB’?

22.  ‘Expression’ term is better to be avoided when speaking about qPCR because an  mRNA level is not always in correspondence with the correspondent protein quantity in a cell.

Author Response

Please see the attachment. High resolution images can be provided if requested.

Reviewer 2 Report

It will be interesting to see DPSCs in different media.

Reviewer 3 Report

Gross T et al. investigated cellular adaptations and corresponding transcriptional changes of DPSCs in response to extracellular stiffness, which provides insights into the mechanobiological responses and cellular adaptations of DPSCs to extracellular elasticity. Overall the manuscript is well-written, and conclusions are convincing. However, here are some major points that need to be improved.

1.     STRO-1 alone could not identify the sorted cells as “stem cells”. According to the classical definition of dental pulp stem cells, the author should characterize the sorted cells by FACS and colony forming unit assay.

2.     Cell morphology in Figure 3.h is different from others in the same group. I suggest the authors change it into a comparable one.

3.     Please split the channels of fluorescent pictures in Figure 3 and 4. When red and green channels are merged, it is hard to compare the expression level.

4.     Since differentiation towards hard tissues is a major problem that could be influenced by extracellular stiffness, I suggest the authors detect the hard tissue differentiation markers with qPCR. Also, in vitro differentiation test results should be presented. 

Author Response

(The authors gave the same response as above.)

Reviewer 4 Report

1. Please discuss the following fact in discussion section

"Expression of periostin was identified in each of the analysed dental pulp cell lines, which can be regulated by TGF-β1 and biomechanical stimulation. Overall, DPSCs are the most responsive cells to stimulation.please discuss the following fact in your discussion sectio "

Round 2

Reviewer 1 Report

Thank you for answering all my questions. I do not have addtional comments and remarks.